# Tinnitus and hyperacusis involve hyperactivity and enhanced connectivity in auditory-limbic-arousal-cerebellar network

Yu-Chen Chen[1], Xiaowei Li[2], Lijie Liu[2], Jian Wang[2,3], Chun-Qiang Lu[1], Ming Yang[1], Yun Jiao[1], Feng-Chao Zang[1], Kelly Radziwon[4], Guang-Di Chen[4], Wei Sun[4], Vijaya Prakash Krishnan Muthaiah[4], Richard Salvi[4]*, Gao-Jun Teng[1]*

[1]Jiangsu Key Laboratory of Molecular Imaging and Functional Imaging, Department of Radiology, Zhongda Hospital, Medical School, Southeast University, Nanjing, China; [2]Department of Physiology, Southeast University, Nanjing, China; [3]School of Human Communication Disorders, Dalhousie University, Halifax, Canada; [4]Center for Hearing and Deafness, University at Buffalo, The State University of New York, Buffalo, United States

*For correspondence: salvi@ buffalo.edu (RS); gjteng@vip.sina. com (GT)

**Abstract** Hearing loss often triggers an inescapable buzz (tinnitus) and causes everyday sounds to become intolerably loud (hyperacusis), but exactly where and how this occurs in the brain is unknown. To identify the neural substrate for these debilitating disorders, we induced both tinnitus and hyperacusis with an ototoxic drug (salicylate) and used behavioral, electrophysiological, and functional magnetic resonance imaging (fMRI) techniques to identify the tinnitus–hyperacusis network. Salicylate depressed the neural output of the cochlea, but vigorously amplified sound-evoked neural responses in the amygdala, medial geniculate, and auditory cortex. Resting-state fMRI revealed hyperactivity in an auditory network composed of inferior colliculus, medial geniculate, and auditory cortex with side branches to cerebellum, amygdala, and reticular formation. Functional connectivity revealed enhanced coupling within the auditory network and segments of the auditory network and cerebellum, reticular formation, amygdala, and hippocampus. A testable model accounting for distress, arousal, and gating of tinnitus and hyperacusis is proposed.

## Introduction

A third of adults over the age of 65 suffer from significant hearing loss, a condition exacerbated by two debilitating condition, subjective tinnitus, a phantom ringing or buzzing sensation, and hyperacusis, normal sounds perceived as intolerably loud or even painful. Roughly 12% of adults experience tinnitus, but the prevalence skyrockets to 50% in young combat personnel (*Leske, 1981*; *Andersson et al., 2002*; *Cave et al., 2007*; *Michikawa et al., 2010*; *Hebert et al., 2013*). Tinnitus is costly with more than $2 billion paid annually in veteran disability payments. Hyperacusis affects roughly 9% of adults (*Andersson et al., 2002*), but its prevalence is likely higher because of the difficulty of self-diagnosis (*Gu et al., 2010*). Remarkably, among those whose primary complaint is hyperacusis, 90% also suffer from tinnitus (*Baguley, 2003*). Since tinnitus and hyperacusis are often triggered by cochlear hearing loss, it was long assumed that these auditory distortions resulted from hyperactivity disorders in the peripheral auditory nerve. This hypothesis, however, is contradicted by studies showing that auditory nerve spontaneous and sound-evoked firing rates are depressed in subjects with cochlear damage (*Kiang et al., 1970*; *Wang et al., 1997*). Moreover, surgical section of the auditory nerve fails to eliminate tinnitus (*Baguley et al., 1992*; *Lockwood et al., 2001*). These negative results plus recent imaging studies now suggest that tinnitus and hyperacusis arise from

**eLife digest** One in three adults over the age of 65 will experience a significant loss of hearing. This is often worsened by related conditions, such as: tinnitus, an unexplained constant buzzing or ringing sound; and hyperacusis, whereby everyday sounds are perceived as too loud or painful.

Most hearing loss is caused by damage to the sound-sensitive cells within a structure in the inner ear called the cochlea. Some studies have also identified regions of the brain that show abnormal activity in people with tinnitus and hyperacusis. However, the results from different patients have often been inconsistent and sometimes contradictory, and so it remains unclear what exactly causes these conditions.

To overcome this problem, Chen et al. made use of the fact that tinnitus and hyperacusis are common short-term side effects of certain drugs and measured the brain activity in rats before and after they were given one such drug. Before receiving the drug, the rats had first been trained to expect to receive a food pellet from the left side of their cage when they heard a steady buzzing sound. The rats were also trained to expect a food pellet from their right if they heard nothing at all. Shortly after receiving the drug, the rats often failed to respond correctly in the 'quiet tests' and behaved like they were already experiencing a constant buzzing sound, as would be expected if they had tinnitus. Further tests confirmed that the drug also triggered behavior in the rats that is typical of people with hyperacusis.

Chen et al. then discovered that the drug treatment reduced the nerve signals that are sent from a rat's cochlea. Moreover, the drug treatment greatly increased the activity in response to sound within parts of the rat's brain; these and other parts of the brain also became overactive in drug-treated rats in the absence of sound. Finally, further experiments revealed that drug-treated rats had stronger connections between these brain regions than in normal rats.

Chen et al. used these results to propose a model to explain the underlying causes of tinnitus and hyperacusis. However, because the drug treatment only induces short-term hearing impairment, further studies are needed to see if this model also applies when these conditions are long-term.

maladaptive neuroplastic change in the central nervous system (CNS) provoked by cochlear pathology (*Lockwood et al., 1998*; *Husain et al., 2011*; *Sereda et al., 2011*).

Several models of tinnitus and hyperacusis have been proposed that involve increased central gain, altered functional connectivity (FC), and aberrant neural oscillations in neural networks (*Weisz et al., 2007*; *Sereda et al., 2011*; *Henry et al., 2014*). Most of these conceptual models have emerged from human imaging studies using magnetoencephalography, electroencephalography, magnetic resonance imaging (MRI), and functional MRI (fMRI) of the blood oxygen level-dependent (BOLD) response (*Llinas et al., 1999*; *Weisz et al., 2005*; *Auer, 2008*; *Gu et al., 2010*; *Moazami-Goudarzi et al., 2010*; *Leaver et al., 2012*; *Maudoux et al., 2012*; *Husain and Schmidt, 2014*). In the context of central gain models, some human imaging data indicate that hyperacusis is associated with enhanced sound-evoked activity in multiple-auditory processing centers, namely auditory cortex (ACx), medial geniculate body (MGB), and inferior colliculus (IC), whereas tinnitus can be triggered solely by enhanced central gain in the ACx (*Gu et al., 2010*). On the other hand, active loudness models suggest that tinnitus arises entirely from increased central noise independent of gain, whereas hyperacusis results exclusively from increased nonlinear gain that results in loudness intolerance (*Zeng, 2013*).

While cross-sectional human brain imaging studies have identified many different sites of aberrant neural activity, published results from patients have often produced diverse, inconsistent, or contradictory findings. Some discrepancies are likely due to confounding factors such as patient heterogeneity, unknown etiology, genetic diversity, social and environmental factors, and duration or severity of tinnitus and hyperacusis. Animal models could potentially overcome many of these limitations provided that tinnitus and hyperacusis can be reliably induced, behaviorally measured, and functionally imaged. While tinnitus can develop in some individuals after intense noise exposure, the percentage of affected individuals is highly variable and its duration is unpredictable (*Heffner and Harrington, 2002*; *Lobarinas et al., 2006*; *Heffner, 2011*). High doses of aspirin, an anti-inflammatory drug used to treat rheumatoid arthritis, have long been known to consistently induce

acute tinnitus in humans and animals (*Myers and Bernstein, 1965*; *Myers et al., 1965*; *Mongan et al., 1973*). Moreover, high-dose sodium salicylate (SS), the active ingredient in aspirin, not only consistently induces tinnitus (*Jastreboff et al., 1988*; *Lobarinas et al., 2004*; *Stolzberg et al., 2013*), but also hyperacusis (*Chen et al., 2014*; *Hayes et al., 2014*); these perceptual disorders disappear a day or two post-treatment. The highly predictable time course of SS-induced tinnitus and hyperacusis makes it an extremely powerful tool for studying the neural correlates of these perceptual disturbances. Therefore, we took advantage of our unique behavioral techniques for assessing SS-induced tinnitus and hyperacusis in rats and combined this with focused electrophysiological measurements plus global fMRI assessment techniques to map out the regions of neural hyperactivity and enhanced FC that characterize the tinnitus–hyperacusis network. To identify regions of heightened or depressed spontaneous neural activity, we measured the amplitude of low-frequency fluctuations (ALFF) in resting-state fMRI (*Zang et al., 2007*; *Zhang et al., 2010*; *Yao et al., 2012*; *Wen et al., 2013*) and combined this with resting-state FC to identify regions of increased or decreased functional coupling between regions of the auditory pathway and other parts of the CNS. This is the first animal study to use ALFF and FC combined with detailed electrophysiological measures to provide a comprehensive neurological map of the tinnitus–hyperacusis network.

## Results

Three complimentary experiments involving behavioral, electrophysiological, and functional imaging were conducted in separate groups of rats. In Experiment 1, three behavioral studies were performed on separate groups of rats to assess SS-induced tinnitus, hyperacusis, and startle reflex hyperactivity. In Experiment 2, electrophysiological measurements were carried out on a separate group of rats to determine how SS altered the neural input/output functions in cochlea, as reflected in the compound action potential (CAP) from the auditory nerve, and the local field potentials (LFP) recorded in the MGB, ACx, and lateral amygdala (LA). In Experiment 3, resting-state fMRI studies were conducted in another group of rats to determine how SS altered the ALFF and FC patterns obtained with seeds placed in ACx, MGB, and IC.

### Experiment 1

#### Tinnitus

To determine if SS-induced tinnitus, we tested three rats using our 2AFC-tinnitus paradigm. All three rats developed tinnitus-like behavior; data from two representative animals are shown in *Figure 1A,B*. During baseline testing (B1–B4; B6–B9), rats correctly identified Quiet (no sound stimulus) trials at greater than 70% correct and AM and NBN trials >80% correct. The saline-control treatment had no noticeable effect on performance during Quiet, NBN, or AM trials. However, when the rats were treated with SS, performance dropped to 50% or less only on Quiet trials, that is, rats shifted their response preference from the feeder associated with Quiet to the feeder associated with a continuous NBN, behavior indicative of tinnitus. When SS treatment was discontinued (P1–P4), performance on Quiet trials returned to baseline indicating that tinnitus had disappeared. Performance on NBN and AM trials was unaffected indicating that behavior was under sound stimulus control.

#### Hyperacusis

To test for SS-induced hyperacusis, we measured reaction time-intensity functions to broadband noise bursts before and after Saline or SS-treatment (*Figure 1C*). Reaction time–time intensity functions obtained with Saline were nearly identical to those obtained during baseline indicating that the injection had no effect on behavior. In contrast, reaction times obtained after SS were significantly different from baseline at low and high intensities [Two-way, repeated measures ANOVA; significant effect of treatment ($p < 0.0001$), intensity($p < 0.0001$), interaction of treatment × sound intensity ($p < 0.001$); Bonferroni post-hoc analysis between baseline and SS significant at 30 dB ($p < 0.001$), 50 dB ($p < 0.05$), 60 dB ($p < 0.01$), 70 and 80 dB ($p < 0.001$), and 90 dB ($p < 0.05$)]. Reaction times after SS were significantly shorter than baseline at moderate to high intensities behavioral evidence indicative of hyperacusis, that is, these intensities were perceived as louder than normal (*Lauer and Dooling, 2007*; *Chen et al., 2014*; *Hayes et al., 2014*). However, at 30 dB SPL (sound pressure level) reaction times were longer than normal due to hearing loss, which reduces the loudness of sounds near threshold. Reaction time–intensity functions returned to normal after SS treatment was discontinued (data not shown).

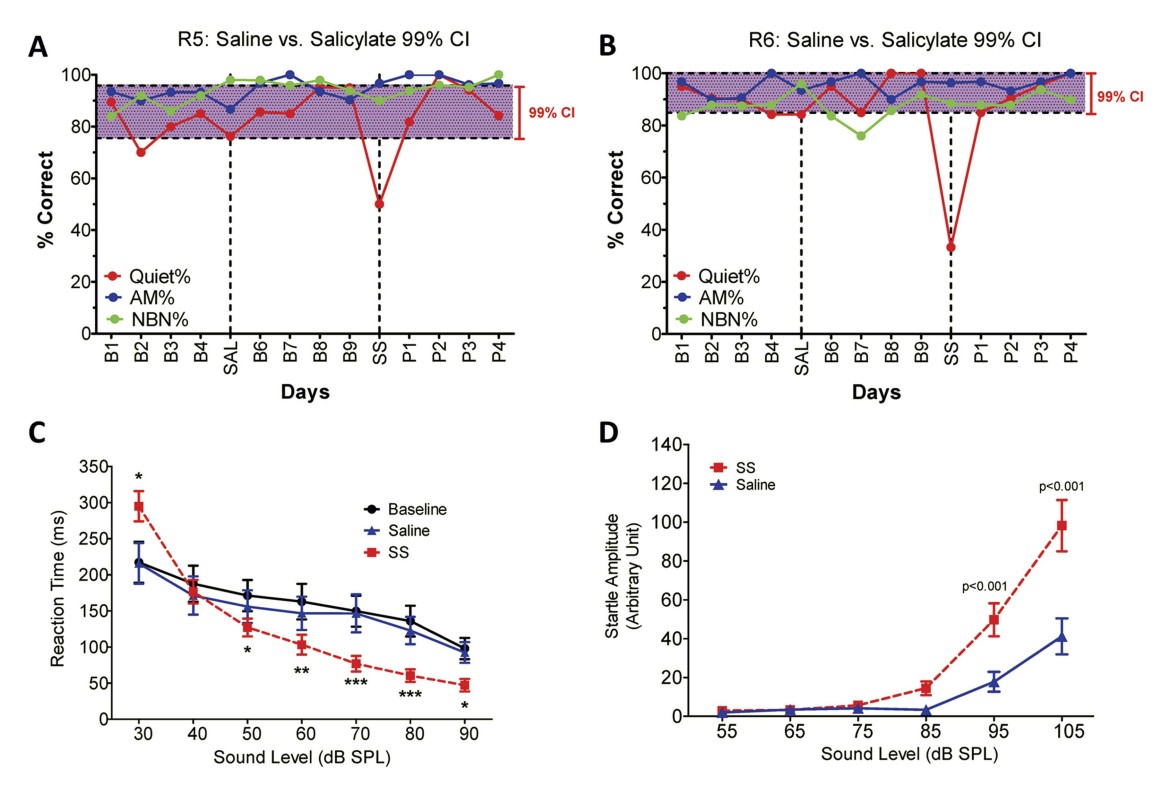

**Figure 1**. SS-induced tinnitus, hyperacusis, and startle reflex hyperactivity. (**A** and **B**) 2AFC-tinnitus task for two representative rats during baseline days B1–B4, during Saline (SAL) treatment, during baseline days B6–B9, 2 hr post-sodium salicylate (SS), and days P1–P4 post-SS treatment. Percent correct performance shown for NBN, AM, and Quiet trials. Purple-shaded region is the 99% confidence interval for baseline measurements (B1–B4; B6–B9). (**C**) Mean (+SEM, n = 7) reaction time-intensity functions measured at baseline, after Saline-treatment and after SS-treatment. Reaction times during SS treatment were significantly longer than baseline at 30 dB SPL and significantly shorter than baseline at 50–90 dB SPL (*p < 0.05; **p < 0.01; and ***p < 0.001). (**D**) Mean (+SEM, n = 6) startle amplitude-intensify functions after treatment with Saline or SS. Startle amplitudes after SS treatment were significantly larger than after Saline at 95 and 105 dB SPL (p < 0.001).

## Startle reflex hyperactivity

Startle reflex hyperactivity has been linked to hyperacusis (*Sun et al., 2009*; *Lu et al., 2011*). Therefore, acoustic reflex amplitude-intensity functions were compared in the same group of rats after Saline and SS treatments (*Figure 1D*). SS treatment caused a significant increase in startle amplitude at 95 and 105 dB SPL [Two-way, repeated measure, ANOVA, significant main effect of intensity (p < 0.001), treatment (p < 0.001), intensity × treatment interaction (p < 0.001); Bonferroni post-test significant at 95 dB and 105 dB (p < 0.001)]. Startle amplitudes returned to normal when SS was discontinued (data not shown).

## Experiment 2

### Electrophysiology

SS is known to cause temporary hearing loss and reduce the neural output of the cochlea. To quantify the effects, CAP amplitude-intensity functions were measured before and 2 hr post-SS. The mean (+SEM) CAP amplitude-intensity function (average of 6, 8, 12, 16, 20, 24, 30, and 40 kHz) measured 2 hr after SS treatment was shifted to the right at low intensities due to a threshold shift of approximately 20 dB (horizontal arrow, *Figure 2A*). In addition, the amplitude of the CAP was greatly reduced (70–80%) at suprathreshold intensities (down arrow, *Figure 2A*) indicating a profound reduction in the neural output of the cochlea. LFP amplitude-intensity functions were also recorded from the MGB, ACx, and LA before and 2 hr after SS treatment. The LFP amplitude-intensity functions from all three structures were shifted to the right approximately 20 dB

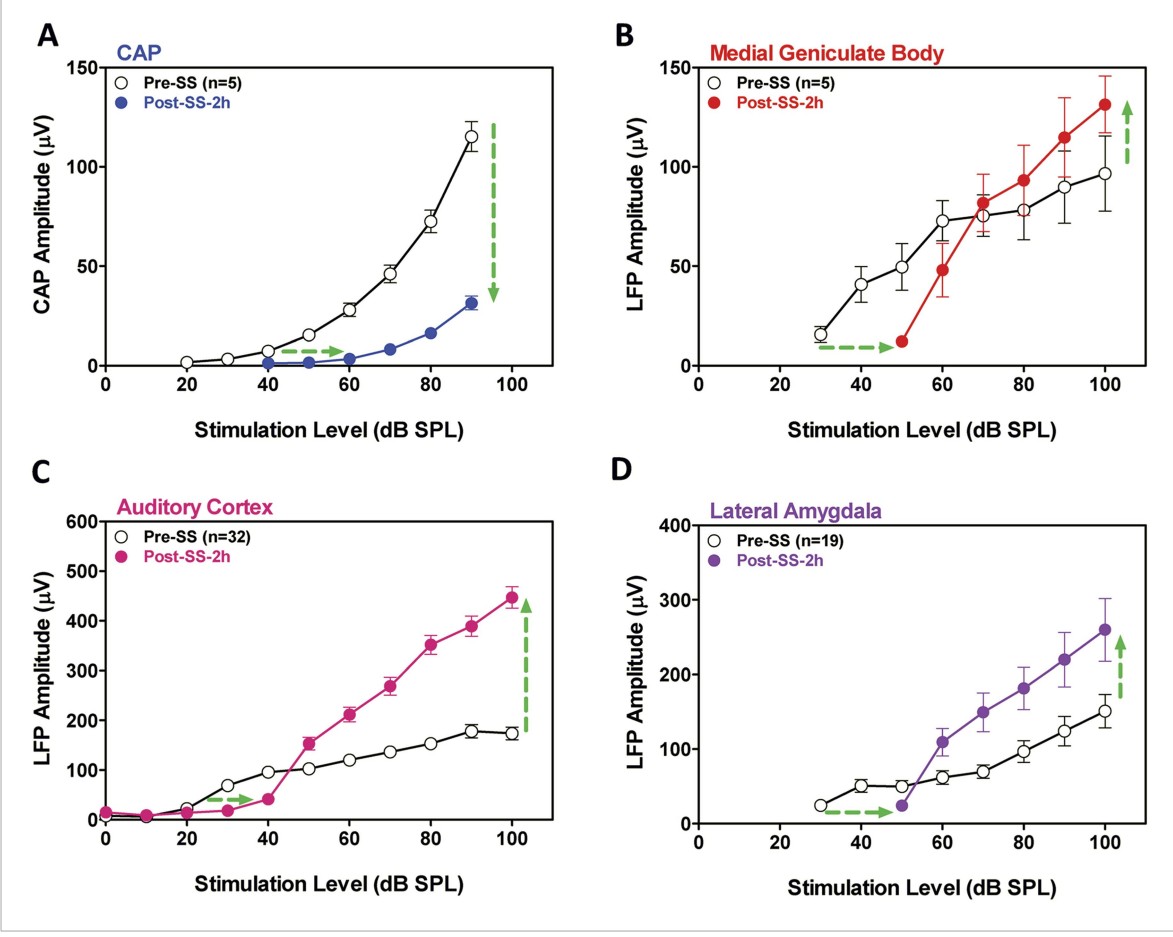

Figure 2. SS depresses cochlear potentials but enhances central auditory evoked responses. Effects of 300 mg/kg SS on peripheral and central electrophysiological measures. (**A**) Mean (+SEM, n = 5) compound action potential (CAP) input/output function (average of 6, 8, 12, 16, 20, 24, 30, and 40 kHz; 10-ms tone burst) recorded from the round window pre- and 2 hr post-SS. Note, 20 dB threshold shift of the function to the right at low intensities (horizontal arrow) and large reduction in CAP amplitude at high intensities (down arrow). (**B**, **C**, **D**) Local field potential input/output functions (50-ms noise bursts) from medial geniculate body, auditory cortex, and lateral amygdala (AMY), respectively, before and 2 hr post-SS. Note, 20 dB threshold shift of the functions to the right at low intensities (horizontal arrows) and large increase in response amplitude (up arrow) at suprathreshold levels (>60 dB SPL).

(*Figure 2B–D*) 2 hr post-SS consistent with the CAP. These results indicate that the threshold shift measured in central structures is largely determined by the loss of sensitivity at the cochlea. LFP amplitudes in the MGB, ACx, and LA increased rapidly with increasing intensity, and response amplitudes became substantially larger than pre-treatment values (*Figure 2B–D*) in contrast to the large CAP amplitude reductions (~70% decrease) (*Figure 2A*). The SS-induced enhancement of suprathreshold LFP amplitudes at 100 dB SPL was approximately 50% in the MGB and 140% in the ACx, results indicative of a progressive increase in gain from peripheral to more central auditory loci (*Noreña, 2010*; *Lu et al., 2011*).

## Experiment 3
### ALFF
To identify the global effects of SS on brain activity, we compared the ALFF in the SS group with the Saline group 2 hr post-treatment using two-sample t-tests corrected for multiple comparisons. *Figure 3* shows the regions where significant increases or decreases in ALFF were observed due to SS; *Table 1* shows the cluster sizes and t-values in left and right hemispheres for each region. Within the cerebellum, SS produced significant bilateral increases in ALFF in the paraflocular lobes (PFL, 38–37 voxels) and cerebellar lobules 4 (CB4, 38–37 voxels) (*Figure 3A,B*). Significant bilateral increases in

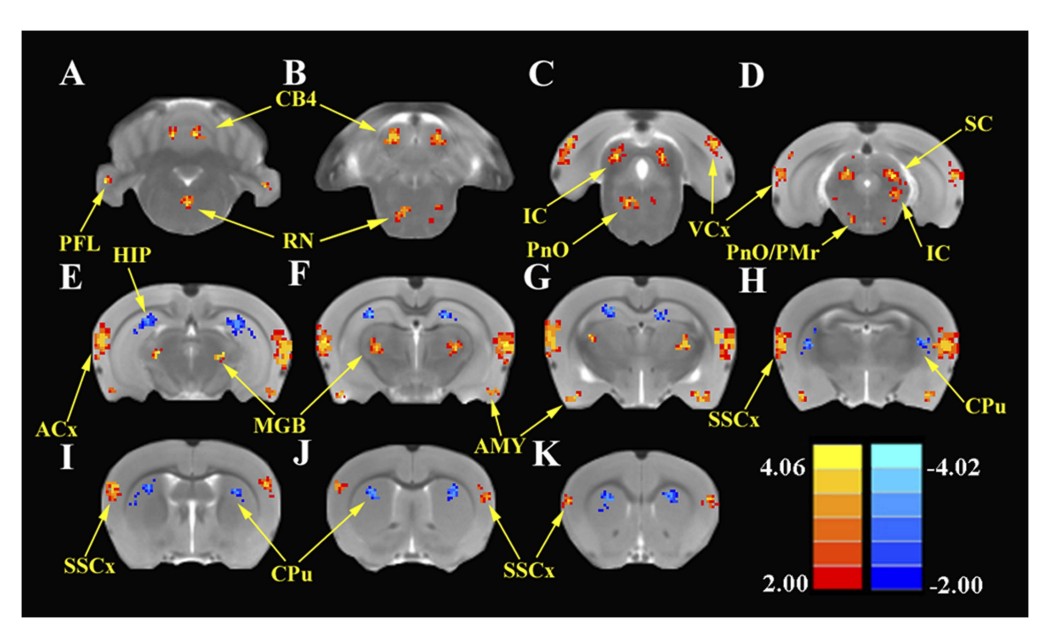

**Figure 3**. SS enhances and depresses amplitude of low-frequency fluctuations (ALFF) in specific CNS regions. Panels **A** (most caudal) through **K** (most rostral) show MR images of rat brain. Significant differences in ALFF between the SS group vs Saline group 2 hr post-treatment. Thresholds set at a corrected p value of <0.001 determined by Monte Carlo simulation. CB4, lobules 4 of cerebellum; PFL, parafloccular lobe of cerebellum; RN, gigantocellular reticular nucleus; PnO, pontine reticular nucleus oral; PMr, paramedian raphe nucleus; VCx, visual cortex; IC, inferior colliculius; SC, superior colliculius; MGB, medial geniculate body; ACx, auditory cortex; HIP, hippocampus; AMY, amygdala; SSCx, somatosensory cortex; Cpu, caudate putamen. Color heat map scale in lower right shows corrected t-values ranging from +4.06 to −4.02.

ALFF were also observed in subcortical areas that included the gigantocellular reticular nucleus/oral region of the pontine reticular nucleus (RN/PnO, 35–28 voxels, *Figure 3C*) and pontine reticular nucleus/paramedian raphe nucleus (PnO/PMr, 12–10 voxels, *Figure 3D*). In the midbrain, significant bilateral increases in ALFF occurred in the IC, a binaural auditory processing area (IC, 72–68 voxels, *Figure 3C,D*) (*Skaliora et al., 2004*) and superior colliculus (SC, 40–43 voxels, *Figure 3D*), a visual center with multisensory inputs (*Wallace et al., 1993*). Significant bilateral increases also occurred in the MGB (MGB, 52–48 voxels), a high-level auditory processing area (*Figure 3E–G*), in the ACx (ACx, 167–178 voxels, *Figure 3E–H*), visual cortex (VCx, 32–39 voxels, *Figure 3C,D*), somatosensory cortex (SSCx, 37–40 voxels, *Figure 3I–K*), and amygdala (AMY, 61–60 voxels, *Figure 3E–H*). In contrast, SS produced significant bilateral decreases in ALFF within the hippocampus (HIP, 108–92 voxels, *Figure 3E–I*) and caudate–putamen (CPu; 72–71 voxels, *Figure 3I–K*).

## Functional connectivity

To determine if SS altered FC, two-sample t-tests were computed to identify regions where significant differences occurred in the FC maps for SS and Saline conditions. As shown in *Figure 4* and *Table 2* (cluster size, t-values shown for left and right hemispheres), when the seed region was in the ACx, there were significant bilateral increases of FC in large clusters located in the MGB (62–70 voxels), IC (82–88 voxels), and AMY (67–68 voxels) plus moderate clusters located in the RN (52–48 voxels), PFL (39–35 voxels), and CB4 (51–53 voxels). No decreases in FC were observed. When the seed regions were located in MGB, there were significant bilateral increases in FC in large clusters in the ACx (195–210 voxels) and the HIP (162–178 voxels). Again, no decreases in FC were seen. Finally, with the seeds located in the IC, there were significant bilateral increases in FC in large clusters located in the MGB (72–60 voxels) and HIP (140–131 voxels); again no decreases in FC occurred. The *Figure 4—figure supplement 1* showed the BOLD data (ALFF and FC) for baseline and after SS application separately.

Table 1. SS-induced changes in amplitude of low-frequency fluctuations (ALFF); SS group vs Saline group; p < 0.001 corrected for multiple comparisons

| Brain region | Left | | Right | |
|---|---|---|---|---|
| | Cluster size | t-value | Cluster size | t-value |
| ALFF increased | | | | |
| ACx | 167 | 3.812 | 178 | 3.746 |
| IC | 72 | 3.383 | 68 | 3.473 |
| MGB | 52 | 3.432 | 48 | 3.339 |
| SSCx | 37 | 3.383 | 40 | 3.402 |
| VCx | 32 | 3.342 | 39 | 3.312 |
| SC | 40 | 4.123 | 43 | 4.094 |
| AMY | 61 | 4.192 | 60 | 3.923 |
| RN/PnO | 35 | 3.249 | 28 | 3.290 |
| PnO/PMr | 12 | 3.498 | 10 | 3.313 |
| PFL | 38 | 3.349 | 37 | 3.292 |
| CB4 | 38 | 3.349 | 37 | 3.292 |
| ALFF decreased | | | | |
| HIP | 108 | −4.087 | 92 | −4.002 |
| CPu | 72 | −3.772 | 71 | −3.741 |

Abbreviations: auditory cortex (ACx), inferior colliculus (IC), medial geniculate body (MGB), somatosensory cortex (SSCx), visual cortex (VCx), superior colliculi (SC), amygdala (AMY), gigantocellular reticular nucleus (RN), pontine reticular nucleus oral (PnO), paramedian raphe nucleus (PMr), parafloccular lobe of cerebellum (PFL), cerebellum lobule 4 (CB4), hippocampus (HIP), caudate-putamen (CPu), sodium salicylate (SS).

## Discussion

### Brain gain

SS induced a peripheral threshold shift of approximately 20 dB for the CAP (*Figure 2A*) (*Chen et al., 2013*, *2014*). The same amount of threshold shift occurred at higher levels of the auditory pathway indicating that the SS-induced hearing loss originates in the cochlea and is relayed centrally. SS also reduced the CAP neural output by ∼70% at suprathreshold intensities. Paradoxically, suprathreshold LFP amplitudes in the MGB, ACx, and LA were larger than normal despite the massive reduction in the output of the cochlea (*Figure 2B–D*). These provocative findings provide compelling evidence for an increase in central gain, a form of homeostatic plasticity implicated in tinnitus and hyperacusis (*Salvi et al., 1990*; *Auerbach et al., 2014*). The enhanced LFPs seen in ACx are consistent with the enhanced fMRI response observed in the ACx of tinnitus patients, whereas the enhanced LFPs seen in both ACx and MGB are consistent with the enhanced fMRI responses observed these regions in hyperacusis patients (*Gu et al., 2010*). These results are consistent with previous models and data linking tinnitus and loudness intolerance to increased central gain in the central auditory pathway in particular regions from the IC to ACx (*Salvi et al., 1990*; *Qiu et al., 2000*; *Auerbach et al., 2014*). In some models, enhanced central gain amplifies central neural noise resulting in tinnitus (*Noreña, 2010*). However, in other models, central neural noise increases independent of central gain (*Zeng, 2013*); this could potentially explain why some patients only experience tinnitus, but not hyperacusis (*Baguley, 2003*). However, this distinction is clouded by the fact that many tinnitus patients are unaware of their mild hyperacusis, that is, hyperacusis may be more prevalent in tinnitus patients than currently believed because many patients are unaware of their hyperacusis (*Gu et al., 2010*).

Many cellular mechanisms could enhance central gain, but one likely candidate is reduced inhibition (disinhibition). Considerable evidence exists for dysregulated inhibition in central gain-control models (*Auerbach et al., 2014*). First, SS can suppress GABA-mediated inhibition and enhance excitability (*Xu et al., 2005*; *Gong et al., 2008*). Second, SS enhances sound evoked activity

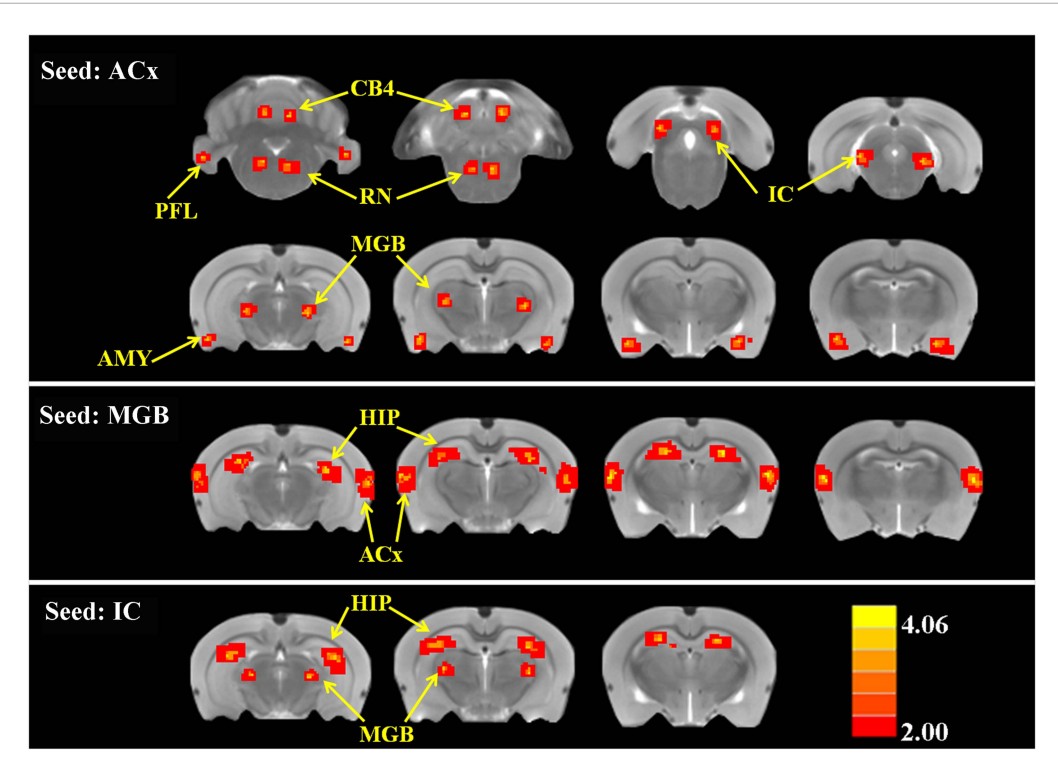

**Figure 4**. SS alters functional connectivity (FC) in specific brain resions. ROI FC heat maps showing the regions of the brain where SS induced a statistically significant increase in FC with the ROI placed in the ACx (top row), MGB (middle row), or inferior colliculus (IC) (bottom row). Scale bar shown in lower right; corrected t-values ranged from +4.06 to −2.00. CB4, lobules 4 of cerebellum; PFL, parafloccular lobe of cerebellum; RN, gigantocellular reticular nucleus; IC, inferior colliculius; MGB, medial geniculate body; ACx, auditory cortex; HIP, hippocampus; AMY, amygdala.

The following figure supplement is available for figure 4:

**Figure supplement 1**. The ALFF and FC data (seeds: ACx, MGB, and IC) for baseline and after SS application separately.

in ACx when given systemically or applied locally to the LA or ACx, whereas it depresses ACx responses when only applied to the cochlea (*Sun et al., 2009*; *Chen et al., 2012*). Third, drugs that enhance GABA-mediated inhibition, prevent SS-induced gain enhancement (*Sun et al., 2009*; *Lu et al., 2011*), and suppress tinnitus (*Brozoski et al., 2007b*).

Behaviorally, hyperacusis was initially observed at 50 dB SPL; the same low intensity at which sound-evoked neural hyperactivity occurred in the ACx. In contrast, sound-evoked hyperactivity occurred at noticeably higher intensities for the LA (~60 dB SPL), MGB (~70 dB SPL), and acoustic-startle reflex amplitude (~95 dB SPL). These results suggest that neural responses from the ACx may be one of the most sensitive biomarkers of hyperacusis (*Juckel et al., 2004*; *Gu et al., 2010*). However, since neural responses increased in magnitude from cochlea to cortex, loudness intolerance issues likely result from multiple stages of neural amplification as signals are relayed rostrally from the cochlea to the ACx (*Auerbach et al., 2014*). Indeed, there is growing evidence that the neural amplification gradually develops in the auditory brainstem and serially accumulates to supernormal levels after reaching the MGB and ACx consistent with previous electrophysiological results (*Qiu et al., 2000*; *Schaette and McAlpine, 2011*).

## Tinnitus

Some models of tinnitus are based on changes in spontaneous spiking patterns such as increased firing rate or increased neural synchrony (*Eggermont, 2015*). SS either decreased or had no effect on

**Table 2**. SS-induced increases in functional connectivity (FC); p < 0.001 corrected for multiple comparisons

| Seed region | Brain region | Left | | Right | |
|---|---|---|---|---|---|
| | | Cluster size | t-value | Cluster size | t-value |
| ACx | MGB | 62 | 3.900 | 70 | 3.859 |
| | IC | 82 | 3.912 | 88 | 3.632 |
| | AMY | 67 | 3.897 | 68 | 3.839 |
| | RN | 52 | 3.983 | 48 | 3.902 |
| | PFL | 39 | 3.992 | 35 | 3.954 |
| | CB4 | 51 | 4.066 | 53 | 4.070 |
| MGB | ACx | 195 | 4.074 | 210 | 4.084 |
| | HIP | 162 | 4.032 | 178 | 4.109 |
| IC | MGB | 72 | 4.098 | 60 | 4.013 |
| | HIP | 140 | 4.064 | 131 | 3.904 |

Abbreviations: medial geniculate body (MGB), inferior colliculus (IC), amygdala (AMY), reticular nucleus (RN), parafloccular lobe of cerebellum (PFL), cerebellum lobule 4 (CB4), auditory cortex (ACx), hippocampus (HIP), sodium salicylate (SS).

spontaneous spike rate in primary ACx (*Ochi and Eggermont, 1996*; *Yang et al., 2007*) and reportedly no effect on synchrony between neuron pairs (*Eggermont, 2015*). Since the BOLD and LFP responses mainly represent presynaptic activity, it is difficult to directly relate our results to these spiking models. However, the increase in very low-frequency BOLD oscillations (0.01 Hz) represented by ALFF could be interpreted as evidence for increased presynaptic synchrony, which would likely enhance spike synchrony albeit at much longer time intervals than previously studied or over much larger neuronal populations than that reflected by spike correlations between neuron pairs. SS has also been found to increase gamma-band (50–100 Hz) oscillatory activity in ACx (*Stolzberg et al., 2013*); oscillations substantially higher than in ALFF. An alternative view is that the tinnitus percept is derived from coordinated activity among several auditory and nonauditory regions (*Horwitz and Braun, 2004*; *Husain et al., 2006*). Enhanced FC between the HIP and auditory areas provides a substrate for assigning a spatial location to a phantom sound, while coordinated activity between specific auditory areas and the reticular formation and AMY may draw attention to and add emotional significance to neural activity in the auditory pathway. Thus, functionally coordinated activity within the network may be essential for bringing tinnitus into consciousness.

## Reallocating network resources

Tinnitus and hyperacusis, like phantom limb pain and cutaneous allodynia, are triggered by peripheral damage presumably leading to widespread changes in the CNS that involve altered connections in networks that include portions of the central auditory pathway and other regions linked to emotion, memory, attention, and arousal (*Llinas et al., 1999*; *Leaver et al., 2012*; *Husain and Schmidt, 2014*). In the resting state, SS increased ALFF and FC in a broad-neural network that included core auditory structures extending from the IC to the ACx consistent with previous studies implicating these central auditory structures in tinnitus (*Paul et al., 2009*). SS also enhanced sound-evoked LFP in the MGB and ACx suggesting a key role for these auditory structures in amplifying auditory information that could manifest as loudness intolerance (*Gu et al., 2010*).

## Cerebellar gating and gain

Although the cerebellum is mainly involved in motor planning and control, some cerebellar regions such as the PFL and vermis receive inputs from auditory centers (*Petacchi et al., 2005*) and respond to sound (*Lockwood et al., 1999*). Interestingly, the perception of tinnitus has been linked to activation of the PFL and vermis (*Brozoski et al., 2007a*) consistent with our results. Since ablation or

inactivation of the PFL eliminates the perception of noise-induced tinnitus (*Bauer et al., 2013*), some have suggested that the PFL acts as a gain control mechanism comparing the afferent input from the cochlea with descending signals from the ACx (*Bauer et al., 2013*). Consistent with this view, our results show that SS leads to hyperactivity in the ACx and increases the FC between the ACx and PFL and CB4. If this cerebellar-tinnitus gating hypothesis is correct, then ablating or inactivating the PFL should suppress behavioral measures of SS-induced tinnitus and possibly hyperacusis, providing a clear test of this model. The functional role of the PFL in tinnitus–hyperacusis network could be further elucidated by inactivating the PFL and determining the effects this has on SS-induced changes we observed in our electrophysiological and fMRI measures.

## Negative valence

The AMY, which assigns emotions such as fear or anxiety to sensory events, lies outside the classical auditory pathway; however, it is linked to several auditory areas and responds robustly to sound (*Romanski and LeDoux, 1993*; *Stutzmann et al., 1998*; *Chen et al., 2014*). In the resting state, SS enhanced the ALFF in the AMY and increased FC between ACx and AMY consistent with prior results showing increased coupling between ACx and AMY in tinnitus patients (*Kim et al., 2012*) and increased c-fos immunolabeling in the AMY following SS treatment (*Wallhäusser-Franke et al., 2003*). SS also enhanced sound-evoked activity in the AMY consistent with the increased activation seen in the AMY of hyperacusis patients (*Levitin et al., 2003*). Importantly, infusion of SS directly into AMY increases sound-evoked activity in the ACx, effects that illustrate the potent independent role that the AMY can exert on central auditory function and aural perception (*Chen et al., 2012*). Collectively, these results reinforce the view that the AMY contributes to the fear and anxiety experienced by many patients with tinnitus and hyperacusis (*van Veen et al., 1998*; *Juris et al., 2013*; *Aazh et al., 2014*). Sound and cognitive therapies aimed at reducing the emotional distress of tinnitus and hyperacusis would be expected to reduce the level of activity in the AMY and/or the functional coupling between the AMY and ACx without necessarily eliminating aberrant auditory percepts (*Hazell and Jastreboff, 1990*). Human imaging studies employing ALFF and FC could be used to test this hypothesis and provide an objective and independent assessment of how these therapies work and their effects on the tinnitus–hyperacusis neural network.

## Arousal

A novel finding observed during resting-state fMRI was the SS-induced enhancement of ALFF in the reticular formation (RN, PnO, and PMr) together with increased FC between the reticular formation and ACx. The reticular formation is an important arousal center with numerous inputs from the cochlear nucleus and IC (*Kandler and Herbert, 1991*). Giant neurons in pontine reticular formation control the amplitude of the acoustic startle reflex (*Koch et al., 1992*), and stimulation of the AMY enhances the response of these giant neurons (*Koch et al., 1992*). Thus, the SS-induced increases of ALFF observed in the AMY and reticular formation likely contribute to the enhancement of the acoustic startle reflex.

## Feeling and seeing

SS unexpectedly increased ALFF in SSCx, VCx, and SC raising the question of whether this might be linked to phantom visual or somatosensory perceptions. However, after an extensive search, we were unable to find evidence of such aberrant somatosensory or visual phenomena. While the heightened ALFF response in these areas is novel, such changes seem reasonable, given the multisensory interactions known to exist between auditory, somatosensory, and visual areas. One possibility is that heightened ALFF activity in ACx, MGB, and IC could spill over and enhance activity in visual and somatosensory areas (*Murray et al., 2005*). However, these increases may not lead to altered perception because FC was not enhanced in visual or somatosensory areas. Alternatively, the SS-induced cochlear loss could unmask pre-existing multisensory circuits in visual and somatosensory areas leading to increased activation (*Barone et al., 2013*).

## Tinnitus–hyperacusis network

Although SS has long been known to cause tinnitus (*Mongan et al., 1973*; *Jastreboff et al., 1988*), it is now clear that it also induces strong hyperacusis-like behavior (*Chen et al., 2014*; *Hayes et al., 2014*).

Although tinnitus and hyperacusis could conceivably arise from different mechanisms (*Zeng, 2013*), they frequently co-occur more frequently than previously believed (*Gu et al., 2010*). Tinnitus and hyperacusis do not exist in isolation but are linked to other brain regions associated with emotions, arousal, memories, spatial location, and motor activity as schematized in the tinnitus–hyperacusis network model defined by our imaging results (*Figure 5*). With the seed region in IC, a significant increase in FC occurred in the MGB; this increase is likely due to the SS-induced enhancement of ALFF in the IC, which is relayed rostrally to the MGB (*Figure 5*, thick black line) resulting in a larger and more coherent MGB response. Similarly, with the seed in the MGB, increased FC occurred in the ACx; this increase is likely due to the increased ALFF and FC occurring in MGB, which is relayed rostrally to the ACx (*Figure 5*, thick black line). With the seed in the ACx, increased FC was seen in the same two lower auditory centers, the MGB and IC, raising the possibility of a recurrent feedback loop in this auditory subnetwork (*Figure 5*, shaded area, bidirectional dashed red lines). These data combined with our electrophysiological results suggest that SS enhances the FC and response magnitude in a central auditory subnetwork that extends from the IC through the MGB to the ACx.

The FC data suggest that the ACx is a major hub in the tinnitus–hyperacusis network with side branches that extend caudally to the AMY, RN, and cerebellum (PFL, CB4); these subdivisions all show large SS-induced increases in ALFF, as well as increased FC with the ACx. The side branch connecting the ACx to the AMY provides a pathway through which emotional significance can be attached to tinnitus or hyperacusis (*Chen et al., 2014*) consistent with earlier studies linking anxiety, annoyance, and fear to tinnitus and hyperacusis (*Moller, 2007*). The ACx-RN network provides a conduit by which increased arousal can increase awareness or attention, enhance motivation, or amplify motor responses to tinnitus or suprathreshold sounds (*Carlson and Willott, 1998*; *Paus, 2000*). The ACx-cerebellar branch could serve as a gating path for tinnitus (*Boyen et al., 2014*) or modulate the motor responses to or perceptual salience of suprathreshold sounds thereby contributing to hyperacusis (*Mobbs et al., 2007*). Increased FC between HIP-MGB and HIP-IC could facilitate the formation or stabilization of a memory trace for tinnitus, assign a spatial location to the phantom sound within or outside the head, or promote the retrieval of previously stored auditory memories (*De Ridder et al., 2006*; *Ulanovsky and Moss, 2008*). Our network model can be explicitly tested by administering a high-dose of SS while activating or inactivating part of the network, such as the PFL, and determining if the manipulation abolishes tinnitus or hyperacusis or alternatively determining how such manipulations affects the electrophysiological and fMRI properties of the network. Since high-dose SS is one of the most predictable and reliable inducers of tinnitus and hyperacusis, SS provides researchers with a powerful tool to gain mechanistic insights into the neurophysiological conditions needed to induce these debilitating auditory perceptions. Since SS-induced tinnitus and hyperacusis are transient phenomena that begin shortly after drug treatment, some of the neuro-pathophysiological changes we observed are likely to be similar to those that occur during the early stages of

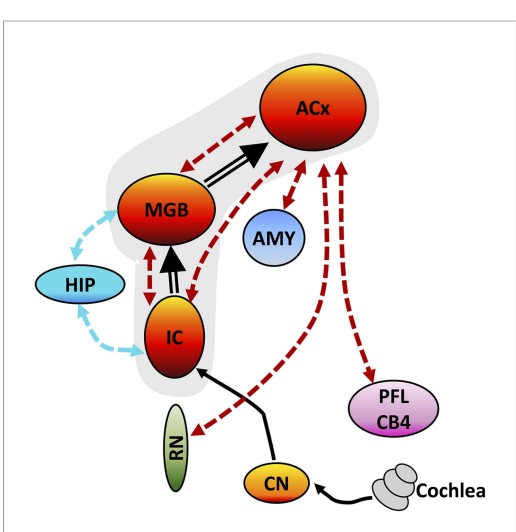

**Figure 5**. Tinnitus–hyperacusis network model. Schematic showing some of the major centers in the auditory pathway and areas in the CNS showing increased FC with the auditory cortex (ACx). Black lines show the neural transmission path from the cochlea through the cochlear nucleus (CN), inferior colliculus (IC), medial geniculate body (MGB), and ACx; black double-line reflects SS-induced increases in ALFF and/or increased FC. SS increased ALFF response magnitudes and FC in a central auditory subnetwork (gray shaded area) comprised of the IC, MGB, and AC. The AC serves as a major hub with side branches to the amygdala (AMY), reticular nuclei (RN), and parafloccular lobe (PFL) and cerebellar lobules 4 (CB4); these side branch regions contribute to the emotional, motoric, and conscious awareness of tinnitus and/or hyperacusis. Enhanced activity in the MGB and IC combined with the increased FC of these auditory structures with the hippocampus (HIP) could facilitate tinnitus memory storage or retrieval or assign spatial location to phantom or real sounds.

tinnitus or hyperacusis induced by acoustic trauma, Meniere's disease, or sudden hearing loss. In cases of acoustic overstimulation, most individuals develop transient tinnitus immediately post-exposure, which gradually disappears; interestingly only a small percentage develops permanent tinnitus (*Gilles et al., 2012*; *Degeest et al., 2014*). Thus, an important question that remains to be answered is whether the neural correlates of chronic tinnitus and hyperacusis are similar or different from the immediate and acute condition.

## Summary

SS-induced tinnitus and hyperacusis appear to arise from enhanced central gain and increased FC in an auditory network with four side branches. The core of the network is composed of auditory structures extending from the IC, through the MGB to the ACx plus branches to the AMY, RN, cerebellum, and HIP. These side branches presumably contribute to the emotional significance, arousal, motor response, gating, and memories associated with tinnitus and hyperacusis.

## Materials and methods

Behavioral Experiment 1 and electrophysiological Experiment 2 were approved by the Institutional Animal Care and Use Committee at the University at Buffalo, Buffalo, NY, USA in accordance with NIH guidelines. The fMRI studies in Experiment 3 were conducted in accordance with the National Institutes of Health Guide for the Care and Use of Laboratory Animals and approved by the Animal Care Committee of Southeast University, Nanjing, China.

## Experiment 1

### SS-induced tinnitus, hyperacusis, and startle reflex hyperactivity

#### Subjects

Sprague Dawley rats were used to obtain behavioral measures of tinnitus, hyperacusis, and acoustic startle reflex amplitude-intensity functions.

#### Acoustic startle reflex

Our procedures for measuring acoustic startle reflex amplitude-intensity functions have been described previously (*Chen et al., 2014*). Six rats were acclimated for 2 days to testing chambers. Acoustic reflex amplitude-intensity functions (50-ms broadband noise bursts, 55–105 dB, 10 dB steps, 18–22 s inter-stimulus interval, 10 presentations/intensity, random order) were measured 2 hr after treatment with Saline (1 ml, i.p.) or SS (300 mg/kg, 1 ml, i.p.). Saline control and SS measurements were separated by 3 days or more, and the order of treatments was randomized across subjects.

#### Hyperacusis assessment

Our procedures for assessing hyperacusis using reaction time-intensity functions are described in detail in recent publications (*Chen et al., 2014*; *Hayes et al., 2014*). Seven food-restricted rats were trained to detect a broadband-noise burst (300 ms, 5 ms rise/fall) presented at intensities from 30 to 90 dB SPL. The rat initiated a trial by holding its nose in a nose-poke hole for 1–4 s until a noise-burst stimulus was presented. If the rat withdrew its nose from the hole during a 2-s response interval that began at noise-burst onset, it received a food pellet, and the trial was scored as a 'hit'. Failure to respond during the 2-s response interval was scored as a 'miss' and not rewarded. Approximately, 30% of trials were catch trials (no stimulus). If the rat removed its nose on a catch trial ('false alarm'), the light in the test chamber was turned off and the rat received a 4-s timeout. However, if it continued to nose-poke through the response interval during a catch trial the trial was scored as a 'correct rejection', and no reinforcement was given. Reaction time measures were only recorded on trials scored as a 'hit'. After obtaining stable baseline reaction time measurements, rats were treated either with SS (300 mg/kg, i.p.; Sigma–Aldrich, St. Louis, MO, USA) or an equivalent volume of Saline and tested on the go/no-go paradigm for approximately 60 min beginning 2 hr post-treatment.

#### Tinnitus assessment

Details of our two-alternative forced choice (2AFC) behavioral paradigm to test for tinnitus are described in previous publications (*Stolzberg et al., 2013*; *Chen et al., 2014*). Three food-restricted

rats (85–90% free feeding weight) were trained to nose-poke a center-hole to start a trial and then wait 4–8 s for a cue light to come on before responding in a bidirectional manner to one of three randomly presented ongoing stimuli. If an unmodulated narrow-band noise (NBN, 50% of trials) was present (4, 5, 6, 7, or 11 kHz center frequency, ~70 dB SPL), the rat was trained to nose-poke the left feeder to obtain a food pellet. In contrast, if an amplitude-modulated (AM, 30% of trials) narrow-band noise (100% modulation depth, 4, 5, 6, 7, or 11 kHz center frequency, ~70 dB SPL) or if no sound (Quiet, 20% of trials) was present, the rat was trained to nose-poke the right feeder to obtain a food pellet. During baseline testing, performance was typically greater than 80% correct. After reaching criterion, rats were treated either with SS or an equivalent volume of Saline and tested daily on the 2AFC paradigm for approximately 75 min beginning 2 hr post-treatment. Since the rats were trained to respond left to a steady NBN vs right to fluctuating AM noise or Quiet, we expected that if a rat developed SS-induced tinnitus, it would only shift its response on Quiet trials from the right side (associated with Quiet) to the left side (associated with a steady NBN) as previously discussed (*Stolzberg et al., 2013*). Behavioral evidence of tinnitus was defined as a significant shift in behavior only on Quiet trials (percent correct below the 99% confidence interval established during baseline testing).

## Experiment 2
### SS-induced changes in auditory electrophysiology

### Subjects

Sprague Dawley rats were used to obtain electrophysiological measures from the cochlea, MGB, ACx, and LA before and after SS treatment (300 mg/kg, i.p.).

### CAP

Rats (n = 5) were anesthetized with ketamine/xylazine (50/6 mg/kg, i.m.), and the CAP recorded before and 2 hr after SS treatment using procedures described in our earlier publications (*Chen et al., 2010*; *Lu et al., 2011*). Tone bursts (6, 8, 12, 16, 20, 24, 30, and 40 kHz, 10-ms duration, 1 ms rise/fall time, cosine gated) were presented at levels ranging from approximately 0 to 90 dB SPL. The response was amplified (1000×), filtered (0.1 Hz–5 kHz), and averaged (50 repetitions). The amplitude of the CAP N1 response was measured and used to construct mean CAP amplitude-intensity functions.

### ACx, MGB, LA

Rats were anesthetized with ketamine/xylazine (50/6 mg/kg, i.m.). LFPs were recorded from the MGB (n = 5), ACx (n = 32), and LA (n = 19) using procedures described in our earlier publications (*Stolzberg et al., 2011*; *Chen et al., 2013*). LFP were recorded before and 2 hr after SS treatment (300 mg/kg, i.p.) using 16-channel silicon electrodes (A-1 × 16–10 mm 100–177, NeuroNexus Technologies). The LFP were filtered (2–300 Hz), sampled at 608 Hz, and averaged over 100 stimulus presentations (50 ms noise-burst, 1 ms rise/fall time, cosine gated). For each intensity, the root mean square amplitude of the LFP was computed over a 50 ms time window for the MGB and ACx and a 100 ms time window for the LA; the data were used to construct LFP amplitude-intensity functions before and 2 hr after SS treatment.

## Experiment 3
### ALFF and FC

### Subjects

Sprague Dawley rats weighing between 180–230 g were used as subjects. Animals were divided into two groups, a Saline-control group (n = 15) and a SS-treated group (n = 15). Rats were anesthetized with urethane (1 mg/kg body weight, i.p.) in order to maintain a stable long-term plane of anesthesia during data acquisition.

### Salicylate

Prior to positioning the rat in the scanner, a catheter (25-gage needle) was inserted into the intraperitoneal space. The catheter was attached to a syringe (5 ml), which contained either normal

Saline or SS dissolved in normal Saline. After collecting all the baseline MRI data, the rats were treated either with normal Saline (3 ml) or 300 mg/kg of SS dissolved in normal Saline (3 ml) (*Lobarinas et al., 2004*; *Stolzberg et al., 2013*; *Chen et al., 2014*).

## MRI acquisition and analysis

Each rat was positioned in the scanner in a prone position. Rectal temperature was maintained at 37.5˚C with a temperature-controlled water blanket beneath the rat. The respiratory rate of the rat was monitored continuously during the entire experiment using an MRI-compatible pulse oximeter. Head position was stabilized with a bite bar and two rods located on opposite sides of the temporal surface of the head.

MRI data were acquired with a 7.0 T animal MRI scanner (PharmaScan, Bruker Biospin GmbH, Germany) using a quadrature surface RF coil. Anatomical images were acquired with a turbo-rapid acquisition relaxation enhancement (RARE) T2-weighted sequence (repetition time (TR)/echo time (TE) = 3200/36 ms, slices = 27, field of view (FOV) = 2.5 × 2.5 cm, number of averages = 1, matrix = 384 × 384, slice thickness/gap = 1/0 mm, flip angle = 90˚). The 27 contiguous anatomical images extended anteriorly from the cerebral-olfactory bulb to the caudal region of the cerebellum posteriorly. The BOLD measurements were acquired with a single-shot gradient-echo echo-planar-imaging (GE-EPI) sequence to acquire multiple slices of images. The parameters were: TR/TE = 2000/19 ms, slices = 27, FOV = 2.5 × 2.5 cm, number of averages = 1, matrix = 96 × 96, slice thickness/gap = 1/0 mm, flip angle = 90˚, 100 volumes. Baseline and salicylate/Saline data acquisition occurred over a period of approximately 2.5 hr. Anatomical and functional scans were obtained from each rat before and 2 hr after administering salicylate or Saline.

## Data processing and statistical analysis

The first 10 time points were eliminated to allow for scanner calibration and adaptation of the subject to the environment. Processing of the fMRI data was carried out with statistical Parametric Mapping software (SPM8, http://www.fil.ion.ucl.ac.uk/spm/) and Resting State fMRI Data Analysis Toolkit V1.8 software (REST, http://www.restfmri.net). Sequential data processing steps included: slice-timing adjustment, realignment and correction for head-motion, spatial normalization to the standard rat brain atlas (*Paxinos and Watson, 2004*), smoothing with an isotropic Gaussian kernel (FWHM = 1 mm), detrending and filtering (0.01–0.1 Hz). Data were excluded if head movements exceeded 1.0 mm of maximum translation in the x, y, or z directions or 2.0 of maximum rotation about the three axes. Images for the ALFF analysis were band-pass filtered (0.01–0.1 Hz). Afterward, a fast Fourier transform was performed on the corrected time series data to obtain the power spectrum in each voxel within the brain; the square root of the power spectrum was calculated to obtain the amplitude at each frequency. To identify significant differences in ALFF values between the SS experimental group and the Saline control group, a between-group, two-sample t-test was calculated. Thresholds were set at a corrected p value of $p < 0.001$ (cluster size > 10 voxels) using the multiple comparison correction obtained with the AlphaSim method employing Monte Carlo simulation. Region of interest (ROI)-based FC analysis was performed for three auditory regions consisting of 9 voxels in the ACx, MGB, and IC. These regions were chosen based on the increased ALFF values observed in these areas during SS and because of their putative roles in tinnitus and hyperacusis. The mean time series for each of these three ROIs was computed for the reference time course. Cross-correlation analysis was then carried out between the mean signal change in each ROI and the time series of every voxel in the whole rat brain. Finally, a Fisher z-transform was applied to improve the normality of the correlation coefficients. Both motion parameters resulting from the realignment and the global signal time course were regressed out during this analysis to improve the specificity of the FC (*Büchel et al., 1996*). For each ROI, two-sample t-tests were performed to identify significant changes in FC between the SS experimental group and the Saline control group. Thresholds were set at a corrected p value of $p < 0.001$ (cluster size > 10 voxels) using the multiple comparison correction obtained with the AlphaSim method employing Monte Carlo simulation.

## Acknowledgements

The fMRI studies were supported by a grant from the National Key Basic Research Program (973 Program) (NOs. 2013CB733800, 2013CB733803), National Natural Science Foundation of China (NOs. 81230034, 81271739), Jiangsu Provincial Special Program of Medical Science (NO. BL2013029), Key Project of Jiangsu Province Natural Science Foundation of China (NO. BK20130577),

Fundamental Research Funds for the Central Universities, and Jiangsu Graduate Student Innovation Grant (NO. KYZZ_0076). The electrophysiological and behavioral studies were supported in part by grants from ONR (N000141210731) and NIH (5R01DC011808). YCC acknowledged the financial support from the China Scholarship Council for his joint PhD scholarship (NO. 201406090139). RS acknowledges support from Overseas Master Project Grant, Chinese Educational Ministry, 2012–17.

## Additional information

### Funding

| Funder | Grant reference | Author |
| --- | --- | --- |
| Ministry of Science and Technology of the People's Republic of China | 2013CB733800 | Gao-Jun Teng |
| Ministry of Science and Technology of the People's Republic of China | 2013CB733803 | Gao-Jun Teng |
| National Natural Science Foundation of China | 81230034 | Gao-Jun Teng |
| National Natural Science Foundation of China | 81271739 | Gao-Jun Teng |
| Natural Science Foundation of Jiangsu Province | BK20130577 | Gao-Jun Teng |
| Fundamental Research Funds for the Central Universities and Jiangsu Graduate Student Innovation Grant | KYZZ_0076 | Gao-Jun Teng |
| National Institutes of Health (NIH) | 5R01DC011808 | Richard Salvi |
| China Scholarship Council | PhD scholarship 201406090139 | Yu-Chen Chen |
| Ministry of Education of the People's Republic of China | Overseas Master Project Grant | Richard Salvi |
| Natural Science Foundation of Jiangsu Province | BK20130577 | Gao-Jun Teng |
| Office of Naval Research | N000141210731 | Richard Salvi |

The funders had no role in study design, data collection and interpretation, or the decision to submit the work for publication.

### Author contributions

Y-CC, Collected the fMRI data, performed the analysis, and wrote and revised the manuscript; XL, LL, Assisted with fMRI data collected and analysis; JW, Helped design and execute parts of the MRI experiment and revise the manuscript; C-QL, MY, YJ, F-CZ, Contributed to the discussion and manuscript revision; KR, Collected, analyzed and helped write and revise the tinnitus and hyperacusis behavioral data; G-DC, Collected the electrophysiological data; WS, VPKM, Collected and analyzed the startle reflex results and write and revise this portion of the manuscript; RS, G-JT, Helped design the MRI experiment and write and revise the manuscript

### Ethics

Animal experimentation: All of the animals were handled according to approved Institutional Animal Care and Use Committee of the University at Buffalo and the Southeast University (Permit Number: HER05080Y) in accordance with the National Institutes of Health Guide. All surgery was performed under ketamine/xylazine anesthesia, and every effort was made to minimize suffering.

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
