## [Decision Letter]

Thank you for sending your work entitled “That Loud Phantom Buzz: Neural Hyperactivity and Enhanced Connectivity in an Auditory-Limbic-Arousal-Cerebellar Network” for consideration at *eLife*. Your article has been favorably evaluated by Timothy Behrens (Senior editor), a Reviewing editor, and two reviewers.

The following individuals responsible for the peer review of your submission have agreed to reveal their identity: Heidi Johansen-Berg (Reviewing editor) and Jos Eggermont (peer reviewer). A further reviewer remains anonymous.

The Reviewing editor and the reviewers discussed their comments before we reached this decision, and the Reviewing editor has assembled the following comments to help you prepare a revised submission.

The reviewers found the paper technically sound, and felt it provided a number of interesting and novel findings but they raised particular concerns about presentation of results and interpretation. The following points would be essential to address in a revised submission.

1) The fact that only transient tinnitus and hyperacusis activity was studied makes the value of these experiments for chronic tinnitus in humans not immediately clear. This should be explained in the paper. The Title (“That loud phantom buzz…”) suggests that the paper is about tinnitus, whereas at most it reflects neural correlates for hyperacusis. The Title should be changed.

2) Throughout the manuscript hyperacusis and tinnitus are interpreted as a similar pathology. There is strong evidence that this may not be true: see e.g. Zeng FG et al., 2012, Hear Res etc. This should be discussed. In addition, it should be made clear throughout which animals (tinnitus or hyperacusis) are being considered. If possible, results should be reported separately for tinnitus and hyperacusis groups for all experiments.

3) The data from experiment 3 are new and exciting. However it would be helpful to see the BOLD data for baseline and after SS application separately. It would also be helpful to see data for tinnitus and hyperacusis animals separately if possible.

4) For BOLD data please clarify how many animals are included in each analysis illustrated (i.e. Figures 3 and 4).

5) Because all activity recorded is presynaptic (LFPs and BOLD) there is no way to relate this to spike recordings in the auditory system. Whereas the LFP and spontaneous BOLD amplitude increases after SS, spike recording in the central auditory system typically decrease. This offers an opportunity for discussion of what constitutes neural correlates of tinnitus. This is missing in the paper. If only presynaptic activity is reflected in these data, then of course the increased spontaneous activity as measured in the BOLD response relates to the evoked LFP activity that suggests increased central gain. The fact that no (compound) spiking activity is recorded needs to be addressed in all their discussions.

---

## [Author Response]

*1) The fact that only transient tinnitus and hyperacusis activity was studied makes the value of these experiments for chronic tinnitus in humans not immediately clear. This should be explained in the paper*.

The Discussion section was modified to point out the implications of transient tinnitus and hyperacusis induced by salicylate and its relevance to chronic tinnitus and/or hyperacusis that can occur in some humans as the result of hearing loss induced by various forms of trauma such as noise exposure, ototoxic drug and aging. The distinction between chronic and acute tinnitus and hyperacusis may not be as clear cut as these terms imply. Chronic tinnitus and hyperacusis sometimes disappears spontaneously or after therapy. On the other hand repeated episodes of acute noise-induced tinnitus can develop into long-lasting tinnitus and/or hyperacusis. One of the most puzzling aspects of tinnitus and hyperacusis induced by noise, aging or ototoxic drugs is why only a small proportion of hearing-impaired humans develop these debilitating symptoms while the majority does not. Salicylate has proved to be a powerful tool for consistently inducing both tinnitus and hyperacusis in humans and animals and for identifying the neural correlates of these disturbing auditory disorders.

*The Title (“That loud phantom buzz…”) suggests that the paper is about tinnitus, whereas at most it reflects neural correlates for hyperacusis. The Title should be changed*.

The title of the paper was modified to reflect the fact that salicylate induced both tinnitus and hyperacusis were being studied. The new title is: “Tinnitus and Hyperacusis Involve Hyperactivity and Enhanced Connectivity in Auditory-Limbic-Arousal-Cerebellar Network”.

*2) Throughout the manuscript hyperacusis and tinnitus are interpreted as a similar pathology. There is strong evidence that this may not be true: see e.g. Zeng FG et al. 2012, Hear Res etc. This should be discussed*.

The Introduction and Discussion sections were modified to take into account differing views regarding the pathologies and neural mechanism that potentially give rise to tinnitus and hyperacusis. The models of Norena and Zeng are discussed along with data indicating that hyperacusis may be under reported in tinnitus patients as noted by Gu et al.

*In addition, it should be made clear throughout which animals (tinnitus or hyperacusis) are being considered. If possible, results should be reported separately for tinnitus and hyperacusis groups for all experiments*.

There appears to be a misunderstanding of how the animals in the behavioral studies of tinnitus and hyperacusis relate to the electrophysiological and fMRI studies. All salicylate-treated rats develop both tinnitus and hyperacusis, not just one of these perceptual disorders. One group of rats was used to document that high dose salicylate induced tinnitus and a second group of rats was used to document that high dose salicylate induces hyperacusis. We have tried to make it clear throughout the revised manuscript that high dose salicylate induces both tinnitus and hyperacusis in all animals. Confusion may have occurred because one group of rats was used to demonstrate that high-dose salicylate reliably induced tinnitus and another group of rats was used to demonstrate that high-dose salicylate reliably induces hyperacusis. We used two separate groups of rats because it would be extremely difficult to train rats on two distinctly different behavioral paradigms. The two groups of behaviorally trained rats were tested at the University at Buffalo in Buffalo, NY, USA. Another group of rats was used to obtain the electrophysiological data at the University at Buffalo. Finally, a fourth group of rats was used for the fMRI experiment conducted at Southeast University in Nanjing, China. The manuscript now clearly identifies these four groups.

*3) The data from experiment 3 are new and exciting. However it would be helpful to see the BOLD data for baseline and after SS application separately. It would also be helpful to see data for tinnitus and hyperacusis animals separately if possible*.

According to your suggestions, we add the BOLD data for baseline and after SS application separately. Please see Figure 4—figure supplement 1. However, as discussed above, high dose salicylate induces both tinnitus and hyperacusis in all animals, not just one of these perceptual disorders. Therefore, it is extremely difficult to acquire the BOLD data for tinnitus and hyperacusis animals separately. We have tried to make it clear throughout the revised manuscript.

*4) For BOLD data please clarify how many animals are included in each analysis illustrated (i.e.*
Figures 3 and 4*)*.

In the original manuscript we clearly stated the number of animals for the fMRI analysis. Animals were divided into two groups, a Saline control group (n = 15) and a SS treated group (n = 15).

*5) Because all activity recorded is presynaptic (LFPs and BOLD) there is no way to relate this to spike recordings in the auditory system. Whereas the LFP and spontaneous BOLD amplitude increases after SS, spike recording in the central auditory system typically decrease. This offers an opportunity for discussion of what constitutes neural correlates of tinnitus. This is missing in the paper. If only presynaptic activity is reflected in these data, then of course the increased spontaneous activity as measured in the BOLD response relates to the evoked LFP activity that suggests increased central gain. The fact that no (compound) spiking activity is recorded needs to be addressed in all their discussions*.

In the revised Discussion we point out that we only measured LFP and BOLD responses, but we discuss the spiking data previously reported by ourselves and others, in particular spontaneous rates and neural synchrony. We provided additional discussion on previous spontaneous spiking models of tinnitus and the potential role that functional connectivity in a neural network may play in bringing tinnitus into consciousness.